# Implementation Outcomes and Challenges of Partnerships between Resource Parents and Parents with Sick Infants in Intensive Neonatal Care Units: A Scoping Review

**DOI:** 10.3390/children9081112

**Published:** 2022-07-25

**Authors:** Sonia Dahan, Claude-Julie Bourque, Catherine Gire, Audrey Reynaud, Barthélémy Tosello

**Affiliations:** 1Faculté de Médecine, Aix Marseille University, CNRS, EFS, ADES, 13915 Marseille, France; barthelemy.tosello@ap-hm.fr; 2Department of Neonatology, Hopital Nord, Assistance Publique Hôpitaux de Marseille, Chemin des Bourrely, CEDEX 20, 13915 Marseille, France; catherine.gire@ap-hm.fr; 3CHU Sainte-Justine Research Center, Montreal, QC H3T 1C5, Canada; claude.julie.bourque@umontreal.ca; 4Faculté de Médecine, Université de Montréal, Montreal, QC H3C 3J7, Canada; 5Responsable, SOS Préma, 14 rue de Longchamp, 92200 Neuilly-sur-Seine, France; audrey.reynaud@sosprema.com

**Keywords:** partnership, NICU, veteran parents, family stakeholders

## Abstract

Parents with a sick child in a neonatal intensive care unit (NICU) usually experience stress, anxiety, and vulnerability. These precarious feelings can affect early parent–child interactions and have consequences for the child’s neurodevelopment. Parents who have had a sick child in an NICU (veteran parents) can offer helpful interventions for these vulnerable families. This article is a scoping review of parental interventions used with the families of NICU infants, and an overview of French perspectives. Two independent reviewers studied the scientific literature published in English between 2001 to 2021 using Covidence software. The databases used were MEDLINE, ISI Web of Science, the Cochrane Database, and Google Scholar. Themes were identified from the articles’ results using an open coding approach. The data are presented in a narrative format. Ten articles were included, and four major themes addressed: (1) description of activities, (2) recommendations, (3) impact, and (4) barriers (resulting from recruitment, training, remuneration, and organization). Activities were very diverse, and a step-by-step implementation was recommended by all authors. Peer-support interventions might be a potential resource for those anxious parents and improve their NICU experiences. These challenges are described by *SOS Préma* in France. This article brings together recent studies on partnership in the NICU. It is an innovative topic in neonatology with vast issues to explore.

## 1. Introduction

Sick children in a neonatal intensive care unit (NICU) are always a challenging experience for their parents [1,2,3]. The literature describes symptoms of stress, anxiety, and post-traumatic stress disorder in mothers of extremely preterm babies [4,5,6]. These symptoms impact the mother’s perception of her child′s vulnerability and their interactions [7]. We know that these early interactions, the attachment process, and the child’s health and development are directly and negatively impacted by a premature birth and subsequent hospital stay [8,9,10].

Moreover, the mothers’ stress and depression can harm her self-confidence as a caretaker for her child, even with basic routines during and after hospitalization [11].

Psychoanalysts and pediatricians, such as Freud, Winnicot, Bowlby, and Brazelton, showed that early interactions and bonding are essential for the child’s neurodevelopment [12], and studies of the impacts of a mother with depression symptoms on the neurodevelopment of her child support this [13,14]. Cognitive neurodevelopment, is enhanced in a 10 year old child who was born premature when the quality of the mother’s response to that child provides improved, appropriate stimulation before the child reaches school age [13]. We also know that the persistence of anxiety and depression symptoms impact the child’s IQ and behavior [13,14].

Therefore, it is expected that interventions addressing the mother’s stress, depression, and anxiety after a premature birth may help improve early interaction between parents and their children, as well as enhance the children’s outcomes [15,16]. Many interventions for parents already exist in NICUs, such as family-integrated care [17,18,19] and neonatal individualized development care NIDCAP [20]. Psychologists and social workers are also very involved and supportive in most neonatal units. 

Interventions offered by veteran parents in NICUs is increasing as a resource. These veteran parents, or resource parents (RPs), have experienced the hospitalization of their own child in an NICU and, from that unique perspective, are able to offer support to NICU families, as well as contribute to clinical, research, and teaching activities [21,22].

Such interventions are expanding in North America, and the literature discussing how to implement partnership with parents is increasing [22,23,24,25,26].

This article is based on a scoping review in which we accessed recent data on current knowledge, recommendations, potential impacts, as well as barriers for implementation of partnership with parents in the NICU. As the literature mainly comes from North America, we included the work of SOS Préma in France, a French association mostly composed of volunteer parents who have had an NICU experience, to offset cultural differences between France and North America.

The main objective of this study was to describe such interventions and their impacts based on scientific sources.

We also describe how SOS Préma in France implements partnerships with parents based on the report of the president of the association.

## 2. Materials and Method

This is a scoping review and a reflection on current experiences and themes from the literature by French NICU providers and the *SOS* Préma team.
Scoping review: A scoping review is an ideal tool for the first step in understanding the literature in an innovative research field [27,28]. As recommended, we followed the PRISMA-sR guide [29] and the JBI’s guidance for a scoping review methodology [29]. This review was then carried out by following five steps: (1) identifying research questions, (2) identifying relevant studies, (3) screening the studies, (4) obtaining data extraction and synthesis, and (5) preparing the presentation of the results.Our review aimed to answer the following questions:(1)How can we implement and develop parents’ interventions in NICUs?(2)Is there a direct impact on the babies’ outcomes from this partnership?(3)What are the barriers for implementation?Identification of relevant studies

Reviewed studies were published from 2011 to 2021, representing a ten-year overview, and corresponded to the growing development of these practices. 

Inclusion criteria: Included articles were peer-reviewed, written in English, and focused on parents’ partnership with healthcare providers in different types of NICU activities: clinical care, peer support, research, administration, and teaching.

Exclusion criteria: Studies of parents’ participation during the hospitalization of their child were excluded, as well as studies about family-integrated care, parents’ perspectives or opinion, bereaved parents, and studies on partnership in pediatric or adult population. Publications before 2010 were also excluded. 

The electronic databases searched were MEDLINE, ISI Web of Science, the Cochrane Database, and Google Scholar. Bibliographies of all selected articles were checked, and the following Medical Subject Headings (MeSH) terms were used: “resource parents”, “veteran parents”, “peer-to-peer”, “peer support groups”, “peer counselling”, “partnership with parents”, “parents’ engagement”, and “family stakeholders”. 

These MeSH terms were combined with: “NICU”, “newborn”, “neonate”, and “preterm babies”. Two reviewers independently selected the articles using Covidence software.

Themes from the selected articles were identified using an open coding approach [30]. 

The first and last author (SD, BT) exhaustively read the articles in their entirety, paying special attention to the description of the partnership interventions, recommendations, impacts, and barriers.
Data presentation

The data from the selected articles are presented in a narrative format. 

## 3. Results

### 3.1. Description and Characteristics of Identified Studies

From t electronic searches, we identified 239 articles, including 7 duplicates that were removed and 100 that were excluded after abstract review. There were 152 full texts assessed for eligibility, 9 of which were included upon agreement between the two independent investigators (Figure 1).

A summary of the characteristics of the nine included articles is presented in Table 1. Of these nine publications, four are qualitative descriptive studies, one is a systematic review, two are randomized control trial studies (one ongoing), and two are prospective mixed-method analyses. These nine publications are from North America.

### 3.2. Identified Themes

After analyzing included studies, four general themes were identified (Table 2): description of partnership activities, recommendations for implementation, impacts, and barriers.

#### 3.2.1. Description of Partnership Activities

The most common activity described and developed by resource parents is peer-to-peer support [23,25]. A large variety of activities are possible [2,21,22,24], such as bedside support, social and cultural activities during hospitalization, hospital design initiatives, quality control projects, integration in committees (clinical and ethical), creating websites, fundraising, testimonies, intervention in the media, and teaching. Bourque et al. proposed a classification based on the resource parents’ direct interactions or involvement with families (research, education, teaching, and administration programs) [21]. This allows institutions to promote and develop partnerships with parents according to their needs and desires, as well as their availability and capacity to participate.

#### 3.2.2. Recommendations: One Step at a Time

Promoting partnerships with parents in an NICU requires time and organization. A step-by-step implementation is recommended by all authors [1,22,25,26].
Creating committees

For a better understanding of and involvement in NICUs, committees should be created with all healthcare team members (physicians, nurses, administrators), along with one or two resource parents. How to recruit these parents is still under debate; it is not possible to currently give guidelines, because it depends on each unit’s culture and interaction between parents and providers. Some authors recommended institutions create a job position dedicated to a partnership program [24]. Other authors feared that employing resource parents may subject them to the employer’s authority and prevent them from freely expressing their values and opinions [21].
Recruitment

Recruitment can be discussed during hospitalization, but the timing between the parents becoming participants and the babies’ hospital discharge is not yet clear. Some activities are more complex and present higher levels of risk [22]. A “pyramid of complexity” is a suggested model that reflects on the tasks’ definition in order to avoid reactivation of a traumatic birth experience. This allows planning a very progressive integration of parents by starting with simpler, easier tasks [22]. Parents with various cultural and linguistic backgrounds should also be recruited to help ethnic minorities and foreign-language-speaking parents during their NICU stay [23].

It is hard to avoid bias during the recruitment of veteran parents. We can imagine that parents from higher social and economic levels would be more comfortable being involved; healthcare providers might also have easier and more fluid interactions with them. In addition, parents who still are in a healing process would be more likely to participate and return to work with the neonatal team, if they perceive this type of involvement as a recuperative and meaningful experience [22].
Remuneration

Remuneration for the resource parents is also debated and may help reduce the bias of recruitment associated with lower-income families. Nevertheless, money should not be the motivation for a partnership, because it implies psychological risks. At minimum, compensation for meals and transportation should always be provided [2,21].
Training and support

Resource parents should receive training from healthcare providers to participate in some activities, especially those involving direct interactions with the birth parents (peer-to-peer), and as well as with simulation projects [22,24].

NICU partnerships with parents is an innovative way to improve care. However, not all risks and barriers are yet known. Thus, authors recommended a team of support for these parents from all caregivers [22].

#### 3.2.3. Impacts

It is hard to measure the quantitative impact of a partnership, as it is always associated with other interventions in an NICU and is already known to have a positive effect on babies’ outcomes. These include Kangaroo care or a newborn individualized developmental care and assessment program (NIDCAP) [33]. An important ongoing randomized controlled trial will measure the Bayley scores of premature babies after peer-support interventions during hospitalization [31].

Nevertheless, we already know that there is an indirect impact with partnerships. The literature describes potential impacts on parents, such as increasing parents’ self-confidence, parents’ capacity to solve problems, adaptative coping, self-esteem, and resilience [21,22,23,24,25,26]. The impact on parents’ emotions were measured in two studies [24,32]. Both studies showed, mostly using qualitative surveys, that parental interventions such as individual peer support or group meetings were appreciated [24,32]. These helped decrease isolation and gave hope [32] to the birth parents, as well as helped to normalize the parents’ emotions and decrease their sadness, guilt, and anger [32]. These indirect impacts might improve parents’ experience in NICUs and facilitate interactions with their child.

The potential impacts on the babies’ environment and development were also described, especially when the resource parents were involved in a quality-improvement interventions [2]. Celenza et al. explained that ensuring *“the clinical environment of the neonate, provides the optimal settings and best possible long-term outcomes for the infant and family”* [2]. 

Finally, we now know that such initiatives are safe and can help resource parents find meaning in their NICU experience, as well as play a role in their own healing process [21,22].

#### 3.2.4. Barriers for Implementation

Barriers and difficulties mainly come from priorities given to technical activities; the neonatal unit’s culture and philosophy; issues in program development, coordination, recruiting, and training time; remuneration; and the unknown outcomes for the babies and their families [2,21,25].

Recommendations suggested in the articles we reviewed are shared to help providers overcome these barriers; however, recommendations from North America might not be applicable in European cultures. It is important for each project team to develop their own approach by considering their resources, culture, and population profile. It is for this reason that information and recommendations from *SOS* Préma were included to reflect the French perspective.

## 4. French Perspectives: *SOS* Préma Actions

Since 2004, the French association *SOS* Préma has sought to provide all premature newborns the following parallel, three-axis approach in order for the infants to have the best opportunity to thrive: 1. support for families, 2. dialogue with medical teams, and 3. raising public authorities’ awareness.

### 4.1. Support for Families

*SOS* Préma has approximately 70 volunteer parents dispersed throughout the country in about sixty local branches. Their main mission is to support parents in hospital neonatal medical units, as well as after the infant’s discharge. Some parents act as moderators for the association on social networks; some answer the association′s toll-free hotline; others participate in updating the informational documents and booklets given to parents during hospitalization. The volunteers are required to sign an agreement with the hospital and the satellite location.

All participants are parents of a child born prematurely or hospitalized at birth. They may become association volunteers after an initial training of approximately six hours and two interviews, including one with the association′s psychologist. Parents also receive ongoing training during an annual two and a half day seminar. These training sessions not only focus on the association itself, but also on the psychological experiences of the parents, the operations of the hospital, fundraising, and communications. Additionally, sessions include scientific aspects such as developmental care or French recommendations of GREEN (*Groupe de Réflexion et d’Évaluation de l’Environnement du Nouveau-né)* [34].

### 4.2. Dialogue with Medical Teams

It is increasingly common for the *SOS* Préma association to be asked at the national level to address users in working groups (such as GREEN of the SFN [Société Française de Néonatologie: Available online: https://www.societe-francaise-neonatalogie.com/ (accessed on 8 June 2022)), as well as being asked to participate at the local level (within perinatal health networks or services). Additionally, the association may participate in research in France or in Europe (initiated by qualified, recognized societies such as the SFN, independent researchers, or even the association itself). The objective is not only to bring the voice of families to professionals, but also to successfully achieve “scientific popularization” for families.

The association also organizes a yearly two-day training for professionals. These sessions concentrate on themes regarding prematurity and hospitalization of newborns.

These are supported by testimonials from volunteer parents and scientifically reinforced by recognized professionals. Finally, the association financially supports projects (via its local branches or the *SOS* Préma Prize), along with dissemination of new practices, by offering necessary equipment throughout France. As an example, in 2019, a donation of 60 transport carriers was offered to facilitate a parental/infant skin-to-skin transfer from the delivery room to the neonatal physician services.

### 4.3. Raising Awareness of Public Authorities

Numerous lobbying actions have been carried out over the past 17 years to enlighten public decision makers regarding the ill-suited circumstances experienced by families during their child’s hospitalization and the child’s school integration. These actions, reinforced by the testimonies of volunteer parents, have made it possible to adapt existing laws or to fill legal voids so that society can better support families with premature infants. These actions include the extension of maternity leave in the event of a premature birth (2006), the organization of a General Assembly on prematurity (2013), the creation of a parliamentary study group on “prematurity and the vulnerable new-born” (2016), and the creation of paternity leave during the hospitalization of a premature newborn (2019). Here again, the testimonies of volunteer parents are essential to raise public awareness.

Engaging parents in *SOS* Préma’s three work areas creates a unique parental bond that enables them to play a key role within the association.

Parental testimony is not contestable, and the strength of these testimonies make the *SOS* Préma training days assert such a forceful impact on professionals. Even though these results are not published, the *SOS* Préma professional training website includes the results of a satisfaction survey completed by professionals after training days with RP [35]. Ninety-nine percent of professionals attest that trainings days with RP had an impact on their perspectives of parents’ experience, and 95% answered that they have a direct impact on their practices [35]. By conveying their experiences with emotion, parents are able to illustrate the issues at hand and influence decision-makers.

Later, each testimony is refined by the *SOS* Préma psychologist to assure that the essentials are being said without modifying the reality of the experience, and that a proper professional distance is always maintained. 

## 5. *SOS* Préma Challenges

The asset of volunteer parents comes with challenges. First, there is a heterogeneity of volunteer parents throughout the country. The association does not “recruit”, and it cannot seek new volunteers when there is a volunteer shortage in a specific locality. This dilemma leaves many hospitals in need of an on-site representative, but with no candidates available to them.

Moreover, many *SOS* Préma activities rely on the availability of a volunteer. The volunteers’ availability is subject to their personal or professional needs for time off; thus, the association cannot impose specific times or expectations in order to meet commitments. As a result, the association employs a team of eight permanent staff, including several parents who have experienced the birth of a premature infant.

This permanent staff is not representative of the population of parents who have had an experience in an NICU. Representation is an issue that is also described in the literature [21].

Another challenge is the cost associated with the volunteer program. The training and equipment of a new volunteer represents a financial investment one hopes will yield benefits for several years. However, over the course of time, volunteers’ interests and availability wane. Therefore, associations need to develop other forms of commitment to “build loyalty”.

Offering new activities to stimulate interest and engagement is an interesting solution, especially for volunteers who have been committed for long periods of time.

## 6. Conclusions

This article brings together recent studies on partnerships in NICUs. It is an innovative field of research in neonatology, and there remains much to explore. To date, no study has described any negative or adverse effects of partnerships with parents. It appears that this intervention is safe for families and for RPs, as long as they are supported by healthcare providers.

Though interventions to help parents in the NICU clearly improve early interactions between parents and infants, the impact on long-term development has not yet been demonstrated, and more studies are needed.

When developing partnerships, the following may occur: 

Systemic barriers, such as a lack of financial resources, lack of time, lack of dedicated human resources to organize an appropriate and ethical partnership with parents as recommended (training, support, remuneration).

Lack of scientific evidence.

Cultural and human barriers. Recruiting parents is not easy and caregivers may not always be ready to work in partnership with and learn from parents. 

Partnerships with parents is a new paradigm for the relationship between parents and providers.

Even if these barriers exist, partnerships with parents in the NICU might be one possible way of improving the quality of care and children’s outcomes.

## Figures and Tables

**Figure 1 children-09-01112-f001:**
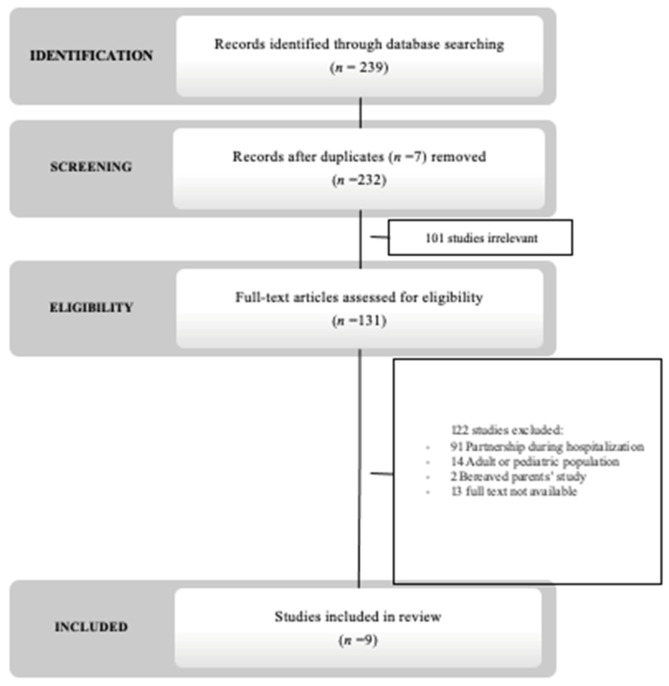
Flow diagram of studies through review process.

**Table 1 children-09-01112-t001:** Characteristics of included articles.

Authors	Year	Location	Study Design	Study Purpose	Participants	Intervention/ Type of Engagement	Measured or Potential Impacts
Ardal et al. [23].	2011	USA	Exploratory qualitative design based on grounded theory	Evaluation of parents’ experience of peer support group with interviews	9 infants, 5 to 20 conversations	Peer support: buddy matching with linguistically and culturally similar parent-buddies	Effect of communication with buddy: being understood/promoting adaptive coping/substitute families and friends/language of emotional support/normalizing effect of shared experience/informational support
Voos et al. [24].	2015	USA	Descriptive qualitative	Description of partnership Impact on emotional support Parent empowerment Welcoming environment Parent education	800 families	Description of partnership: donation of materials, toys, books Revision and translation of handouts Fundraising Parent education for NICU staff on: communication, family in crisis, father’s perspective Parents involved in QI ** (family hand hygiene with videos), NICU committees	A few quotes saying that these initiatives are appreciated
Hall et al. [25].	2015	USA	Systematic review	Description of peer-to-peer support groups and their impacts		Peer-to-peer support in NICU -telephone -in person (best practice) -support groups -internet support group	Potential impacts known in the literature: Increasing parents ‘confidence parents’ capacity to solve problems adaptative coping Perception of social support Self esteem Resilience
Celenza et al. [2]	2017	USA	Descriptive qualitative	Family involvement in QI		Participation of RP * in writing recommendation for QI	Recommendations for developmentally supportive environment, and psychologically supportive for families
Bourque et al. [21]	2018	Canada	Descriptive qualitative 2 centers	Definition of RP Description and classification of activities		RP participation in clinical, administration, research, or teaching activities	Not evaluated in this article
Carty et al. [31]	2018	USA	Ongoing RCT	Measuring impacts of peer-to-peer support intervention	300 parents–infants	RP intervention: formal needs assessment (emotional, personal, financial, and equipment-related) was conducted at start of the navigator–parent relationship and during monthly follow-ups	Increase self-efficacity Decrease stress overall anxiety and depression number of rehospitalizations. Increase vaccination/Bayley score
Dahan et al. [22]	2019	Canada	Prospective qualitative mixed-method analysis	Analyze activities involving parents and exploring RPs’ perspectives and providers opinion	30 RP	653 activities	Self-perceived impact on RP and on providers who worked with them (surveys with closed and open ended questions)
Dahan et al. [26]	2020	Canada	Prospective pilot study mixed-method analysis	Describe the creation and development of a peer-to-peer support meeting	61 parents participated 24 answered questionnaire	Peer-to-peer support meetings in the NICU	79% = meeting very useful
Dahan et al. [32]	2021	Canada	Prospective mixed-method study	Evaluation of parents’ perspectives about peer-to-peer support meetings	45 parents participated, 43 answered the survey	Peer-to-peer support meetings in the NICU	Meeting useful (95%), decreasing isolation (73%), giving hope (63%), getting practical information (32%), normalizing emotions by sharing with others (92%) Decreasing guilt, sadness, anger Improving communication with family and providers

* RP = resource parents; ** QI = quality improvement.

**Table 2 children-09-01112-t002:** Themes generated from the analysis of included studies.

Themes	Examples
Description of activities	“653 activities in 47 types of initiatives” [22] “Types of peer support” [25]
Recommendations	“Partnering with resource parents and patients: practice points and recommendations” [22] “Communicating with others, being a team, starting small, preparing the team, obtaining feedback from providers and parents, documentation and measuring impact, creating a community” [21] “Framework of family involvement in quality improvement…requirements to establish level V family advisor involvement” [2] “Volunteer training: (1) defining the roles of a parent mentor (2) defining the expectations of a parent mentor (3) development of essential skills set” [2]
Impacts	“Improving care/making a difference…giving back/helping other parents…meaning making” [29] “Reporting impacts of meetings on NICU parents: impacts on negative feelings…impacts on communication…impacts on parents’ self-confidence with babies’ care” [32]
Barriers	“Failure to establish this as the goal of neonatal intensive care limits the contribution the dedicated professional team is able to make” [2] “Even when peer support programs are offered…many families still encounter barriers to accessing them. Each family needs may vary, making it difficult for a peer support program to provide a best fit” [25]

## Data Availability

Data are available in articles selected in this review.

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
