# Peer review of "Implementation Outcomes and Challenges of Partnerships between Resource Parents and Parents with Sick Infants in Intensive Neonatal Care Units: A Scoping Review"

_children, 2022, doi:10.3390/children9081112_

Round 1
Reviewer 1 Report
This paper is a scoping review on a very important topic: peer support in neonatology. It is interesting. However I have several major and minor comments.
Abstract: The abstract is clear. However, I suggest the sentence: “The authors’guidelines might be a potential resource for those anxious parents…” should be reformulate. In fact, it is not the author’s guidelines which are a resource, it is the peer support.
Introduction:
The introduction is globally clear.
The reference 13 is not on biology of neurodevelopment.
The reference 15 is on socio economics inequalities in very preterm birth.
I don’t agree with the main objective of the study: It is not to promote such interventions, but to describe these intervention and their results. The collaboration with SOS Prema is potentially interesting but is not evidence based.
Methods:
In the key words, I suggest “peer counselling” should be added.
Results:
Paragraph 3.2.1 Last lines: “Activities involving bereaved parents… families” appears twice, and in the methods (identification of relevant studies) it is written that studies about bereaved parents were excluded.
Table 1: I don’t understand the rationale for including reference 32: in this study, infants hospitalized in NICU are excluded.
For reference 26, no description of intervention and its impact is given.
For reference 22, no description of the intervention is given, no description of the impact is given.
For reference 33, I suggest parent navigator intervention should be changed to peer to peer support, which is the usual term in the current paper. The term number of hospitalization should be changed by rehospitalization for clarty.
Table 2, Barriers “failure to establish.. of neonatal” neonatal appears twice.
French perspectives:
Line 131:” it is an absolute truth” This assertion should be scientifically supported by references.
SOS Préma Challenge
The representability of the resource parents is also a challenge, as the availability of the birth parents for a peer to peer support.
Conclusion
Last sentence: … seems to be the direction… This should be qualify as it is not the only way.
Reviewer 2 Report
It is very interesting manuscript with very important subject.
Manuscript is very good organized, the most important topics are stressed out, appropriate methods are used for literature review.
It will be good to add, to make more visible - exclusion and inclusion criteria.
I don't have any comment to add.
Good luck.
Author Response
We made Inclusion and exclusion criteria more visible.
- Identification of relevant studies
Reviewed studies were published from 2011 to 2021, representing a ten-year overview, and corresponding to the growing development of these practices.
Inclusion criteria: included articles were peer-reviewed, written in English, and focused on parents’ partnership with healthcare providers in different types of NICU activities: clinical care, peer support, research, administration, teaching.
Exclusion criteria: studies on parents’ participation during the hospitalization of their child were excluded as well as studies about
family-integrated care, parents’ perspectives or opinion, and bereaved parents, studies on partnership in pediatric or adult population. Publications before 2010 were also excluded.
Round 2
Reviewer 1 Report
The manuscript is now improved. I would only suggest that in table 1, the reference 31, the ressource parent intervention coud be briefly described.
Author Response
Please find attached the revised manuscript.
This manuscript is a resubmission of an earlier submission. The following is a list of the peer review reports and author responses from that submission.
Round 1
Reviewer 1 Report
The paper is interesting as it reviews literature on peer support in NICU where parents who have had NICU infants will mentors others going through similar experiences.
First of all, the title was hard to grasp, considering the term Resource parent isn't something one hears everyday it was hard to tell what they want. The authors need to define resource parent earlier in the introduction not at the end so that readers understand upfront what exactly they are talking about
The whole introduction also largely focuses on stressors of parents in the NICU, while this is important they needed to introduce a little about their phenomenon of interest and why it is important.
Page 2 like 70 should read review not revue.
The methods section lacks depth. Is this a narrative review where you randomly discuss literature or were you intending to do a scoping or systematic review since you utilized covidence as your software I am assuming you were doing either a scoping or systematic review. If yes then consider the following
- was this review registered with Prospero? If no why?
- how did you decide on 2010?
- where is the search strategy?
- eligibility criteria for inclusion and exclusion for the articles?
- risk bias assessment?
- who did the reviews? from screening to extraction
- on characteristics of studies included - where were these studies conducted? who were their sample/population from which the data was drawn?
- Use PRISMA 2020 guidelines including for the Prisma diagram
- how was the data synthesized?
There are so many missing pieces in the methods section that threatens the validity of this review.
The results and discussion also need reviews to align clearly with the purpose of the review or revise your purpose so you capture or you need (another way is to discuss the other findings as secondary outcomes)
Reviewer 2 Report
This review intended to summarise studies on parent involvement/resource parents in the NICU. This is a study with potential, however, the submitted manuscript doesn't look like a finalised version and a lot of work need to be done to complete the work.
- The abstract should be a total of about 200 words maximum.
- The introduction should offer more background information including an explanation of the key terminologies/ concepts. Try to avoid very short paragraphs without a whole message to deliver.
- the method used in the review was very hard to follow. A detailed searching protocol is needed, searching strings should be submitted in the appendix. I'd like to see detailed inclusive and exclusive criteria when selecting the studies, especially considering the big variations in the relevant studies/methods used in those studies.
- The result section was difficult to follow either, but how the recommendations were generated were particularly hard to understand. The flow chart did not look finished.
- There were some really interesting points in the discussion, but again, it was not written in a professional/logical way, with spelling errors, odd spaces, etc. everywhere.
In general, the manuscript looks more like a working document that is not really ready for publication yet.